# Synthesis and Design of Hybrid Metalloporphyrin Polymers Based on Palladium (II) and Copper (II) Cations and Axial Complexes of Pyridyl-Substituted Sn(IV)Porphyrins with Octopamine

**DOI:** 10.3390/polym15041055

**Published:** 2023-02-20

**Authors:** Anastasia E. Likhonina, Galina M. Mamardashvili, Ilya A. Khodov, Nugzar Z. Mamardashvili

**Affiliations:** G.A. Krestov Institute of Solution Chemistry of the Russian Academy of Sciences, Akademicheskaya St.1P, 153045 Ivanovo, Russia

**Keywords:** Sn(IV)porphyrin extra-complexes, porphyrin polymers, highly porous materials, photoresistance, thermal stability

## Abstract

Supramolecular metalloporphyrin polymers formed by binding tetrapyrrolic macrocycle peripheral nitrogen atoms to Pd(II) cations and Sn(IV)porphyrins extra-ligands reaction centers to Cu(II) cations were obtained and identified. The structure and the formation mechanism of obtained hydrophobic Sn(IV)-porphyrin oligomers and polymers in solution were established, and their resistance to UV radiation and changes in solution temperature was studied. It was shown that the investigated polyporphyrin nanostructures are porous materials with predominance cylindrical mesopores. Density functional theory (DFT) was used to geometrically optimize the experimentally obtained supramolecular porphyrin polymers. The sizes of unit cells in porphyrin tubular structures were determined and coincided with the experimental data. The results obtained can be used to create highly porous materials for separation, storage, transportation, and controlled release of substrates of different nature, including highly volatile, explosive, and toxic gases.

## 1. Introduction

Coordination polymers based on metal complexes of tetrapyrrole macrocyclic compounds represent a special group of self-organized systems that combine their structural fragments through non-covalent interactions [1,2,3,4,5,6,7,8,9,10,11,12,13,14,15,16,17,18,19,20,21,22,23,24,25,26]. Such interactions are reversible and easily controlled by various external influences. Therefore, the coordination self-assembly is a convenient tool to create stable and controllable architectures by the self-organization of metal and organic components with electronic and geometric complementarity. Different functional substituents, the type of organic fragments binding, the metal nature, and the length of the linking fragments provide a variety of framework structure types and areas of their potential application [1,2,3,4,5,6,7,8,9,10,11,12,13,14,15,16,17,18,19,20,21,22,23,24,25,26,27,28,29,30,31,32,33,34,35,36,37]. The construction of supramolecular porphyrin scaffolds is still little studied, but it is undoubtedly an urgent scientific problem, the solution of which will to get closer to understanding the life processes of nature.

In according to our interests in the development of polyporphyrin arrays with controlled practically useful functional properties [24], the synthesis and design of new supramolecular porphyrin oligomers and polymers obtained using reaction of chelation of Sn(IV)porphyrin diaxial complexes with Cu(II) and Pd(II) cations are described in this work. It was shown that the variation of the metal cations nature, and of the flexibility/rigidity of porphyrin macrocycles will give wide range of practically useful properties of these materials. The results obtained can be used to create highly porous materials for separation, storage, transportation, and controlled release of substrates of different nature, including highly volatile, explosive, and toxic gases.

## 2. Experimental

### 2.1. Equipment

All the ^1^H NMR (500.17) experiments were performed on a Bruker Avance III 500 NMR spectrometer with 256 or 512 scans and spectral windows of 20 ppm. The inaccuracy of the ^1^H NMR chemical shift measurement relative to the solvent (CDCl_3_ and DMSO) was found to be ±0.01 ppm. The mass spectra were obtained on a SolariX XR Ion Cyclotron Resonance Mass Spectrometers (Bruker, USA). Elemental analyses were performed on a CHN analyzer Flash EA 1112. The UV–Vis spectra were recorded on a Cary 300 spectrophotometer (Agilent, Santa Clara, USA). The fluorescence spectra were obtained on an RF 5301PC Spectrofluorimeter (Shimadzu, Duisburg, Germany). The porosity of the samples was studied on a NOVAtouch NT LX specific surface and porosity analyzer (Quantachrome, Florida, USA). X-ray diffraction data were obtained using a D2 PHASER diffractometer (Bruker, Cambridge, UK).

### 2.2. Computational Details

The density functional theory (DFT) calculations were carried out using the GAUSSIAN 16 quantum chemical program package [38]. The geometry optimization, the vibrational frequency calculations and the analysis of the potential function of internal rotation of molecules in the ground state were performed using the CAM-B3LYP functional [39] in combination with the def2-TZVP [40] basis set for Sn(IV) and 6-31G* [41] basis set for all other non-metallic atoms. The time-dependent density functional theory (TD-DFT) calculation was performed for geometry-optimized structure at the same model chemistry. The solvent effect (DMF) in TD-DFT calculations was taken into an account by applying the polarizable continuum model (PCM) [42]. The results of the quantum chemical calculations were visualized in ChemCraftsoftware [43].

### 2.3. Synthesis



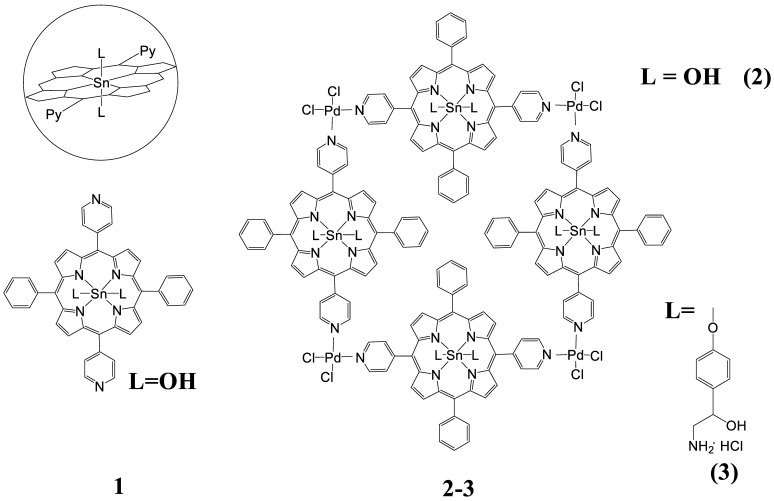





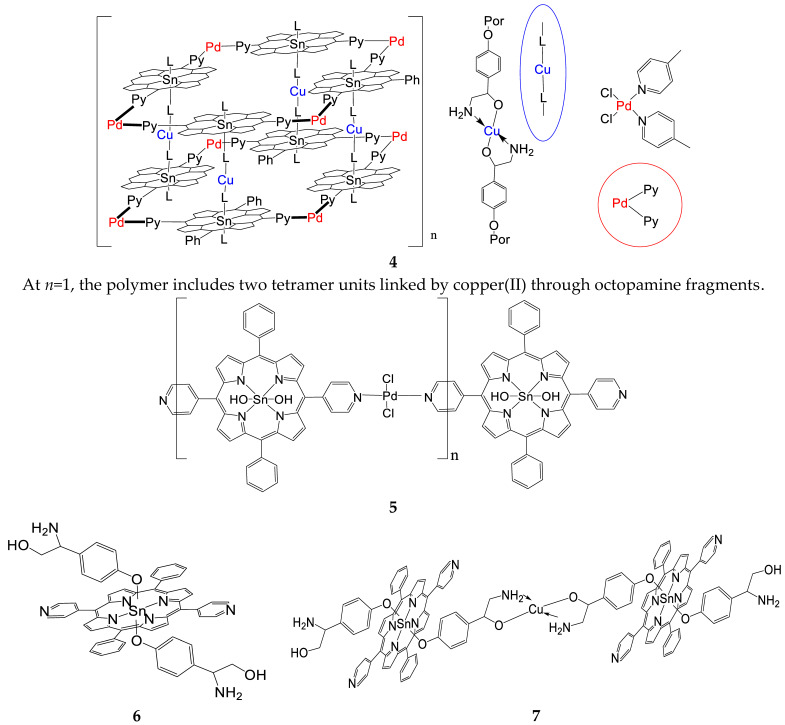



Commercially available 5,15-diphenyl-10,20-di(4-pyridyl)-21H,23H-porphine from PorphyChem, PdCl_2_(C_6_H_5_CN)_2_from Alfa Aesar, and p-octopamine (p-hydroxyphenylethanolamine) from Sigma-Aldrich were used.

Synthesis and spectral characteristics of *dihydroxy*-Sn(IV)-5,15-diphenyl-10,20-di(4-pyridyl)porphyrin (Sn(OH)_2_P, **1**): 20 mg (0.0324 mmol) of 5,15-diphenyl-10,20-di(4-pyridyl)-21H,23H-porphine and 61.4 mg (0.324 mmol) of SnCl_2_ was dissolved in 20 mL of DMF, and refluxed for 1 h. The reaction mixture was cooled down, then purified by column chromatography with aluminum oxide as the adsorbent, and CH_2_Cl_2_ and CHCl_3_ as the eluents. The yield was 13 mg (52%). ^1^H NMR (500 MHz, CDCl_3_): 9.28. d (4H, *β*-H), 9.22 d (4H, *β*-H), 9.13 m (4H, *o*-Py), 8.34 d (4H, *m*-Py), 8.32 d (4H, *o*-Ph), 7.89 m (6H *m,p*-Ph), −7.39 (2H, OH). δ ^13^C ЯMP (DMSO-*d*6): δ 148.6, 135.0, 133.3, 129.6, 127.2. IR (KBr, cm-1): ν *OH* 3590. UV-vis (DMSO, nm): λ*_max_*(logε) 602 (4.26), 562 (4.43), 426 (5.68), 405 (4.84), and MS (ESI): *m*/*z* 769.5059 [Calculated: (*M*)^+^ 767.4298]

Synthesis and spectral characteristics of tetramer based on *dihydroxy*-Sn(IV)-5,15-diphenyl-10,20-di(4-pyridyl)porphyrin with PdCl_2_(C_6_H_5_CN)_2_ salts {(Sn(OH)_2_P)_4_(PdCl_2_)_4,_**2**}: 5.35 mg (6.97 µmol) of Sn(OH)_2_P was mixed with 3.1 mg (8.08 µmol) (C_6_H_5_CN)_2_PdCl_2_, and dissolved in 18 mL of DMF. The reaction mixture was boiled for 3 h, then cooled, diluted with water, and the product extracted into chloroform. Then, the chloroform solution was evaporated, the substance was dried. The yield was 4 mg (60%). ^1^H NMR (500 MHz, CDCl_3_): 9.30 (d, 4H, *o*-Py), 9.20 (d, 4H, *β*-H), 9.16 (d, 4H, *β*-H), 8.33 (d, 4H, *m*-Py), 8.32 (d, 4H, *o*-Ph), 7.93 (tr, 2H, *p*-Ph), and 7.87 (tr, 4H, *m*-Ph). UV- vis (DMF, nm): λ*_max_* 599, 559, 520, 424, and 404.

Synthesis and spectral characteristics of the tetramer **2** with octopamine molecules {(Sn(L)_2_P)_4_(PdCl_2_)_4,_**3**}: 2 mg (0.53 µmol) of tetramer **2** was mixed with 0.81 mg (4.24 µmol) of octopamine hydrochloride and dissolved in 10 mL of DMF. The obtained solution was refluxed for 3 h. The solvent was then evaporated, the resulting solid was washed of water and dried. The product yield was 2 mg (74%). UV-vis (DMF, nm): λ*_max_*, 628, 598, 559, 514, 425, and 404. ^1^H NMR (500 MHz, CDCl_3_): 9.32. d (4H, *o*-Py), 9.23 d (4H, *β*-H), 9.17 d (4H, *β*-H), 8.35 d (4H, *m*-Py), 8.33 d (4H, *o*-Ph), 7.95 tr (2H, *p*-Ph), 7.88 tr (4H, *m*-Ph), 6.13 (s, br., 4H, NH_2_ (L)),5.88 (d, *J* = 8.0 Hz, 4H, *m*-Ph (L)), 5.13 (s, br.,2H, OH(L)), 4.10 (s, 2H, -CH-(L)), 3.11 (t., *J* = 2.0 Hz, 4H, -CH_2_-(L)), 1.90 (d, *J* = 8.0 Hz, and 4H, *o*-Ph (L)).

Synthesis of a polymer based on (Sn(L)_2_P)_4_(PdCl_2_)_4_ bonded by copper (II) salts {((Sn(L-Cu-)_2_P)_4_(PdCl_2_)_4_)_n,_**4**}: 2 mg (0.39 µmol) of tetramer with octopamine **3** was mixed with 1.2 mg (7.87 µmol) of copper(II) chloride dihydrate and dissolved in 15 mL of DMF. The reaction mixture was refluxed for 2 h. Then, the reaction mixture was repeatedly extracted by adding chloroform and water. The chloroform layer was separated, the solvent was evaporated. The obtained solid was washed with water and dried. The product yield was 2 mg (53%). UV-vis (DMF, nm): λ_max_, 598, 557, 425, and 404. ^1^H NMR (500 MHz, DMSO-d_6_): 9.31. d (4H, *o*-Py), 9.23 d (4H, *β*-H), 9.18 d (4H, *β*-H), 8.35 d (4H, *m*-Py), 8.33 d (4H, *o*-Ph), 7.95 tr (2H, *p*-Ph), 7.88 tr (4H, *m*-Ph), 6.11 (s, 2H, NH_2_-L*(→Cu)*), 6.62 (s, 2H, NH_2_-L*(→Cu)*), 5.89 (d, *J* = 8.0 Hz, 4H, *m*-Ph (L)), 5.13 (s, br.,2H, OH(L)), 4.05 (s, 2H, -CH-(L)), 3.10 (t., *J* = 2.0 Hz, 4H, -CH_2_-(L)), and 1.89 (d, *J* = 8.0 Hz, 4H, *o*-Ph (L)).

Synthesis and spectral characteristics of the triad consisting of Sn(IV)-5,15-diphenyl-10,20-di(4-pyridyl)porphyrin and two molecules of octopamine (Sn(L)_2_P, **6**): 11.65 mg (15.2 µmol) of Sn(OH)_2_P **1** was mixed with 6.3 mg (41 µmol) of octopamine hydrochloride and dissolved in 7 mL of DMF. The reaction mixture was refluxed for 2 h. Then, the reaction mixture was repeatedly extracted by adding chloroform and water. The chloroform layer was separated, the solvent was evaporated. The resulting solid was washed with water and dried. The product yield was 11 mg (66 %). ^1^H NMR (500 MHz, DMSO-*d*_6_): 9.28. (m, 8H, *β*-H), 9.22 (d, 4H, *β*-H), 9.13 (m, 4H, *o*-Py), 8.34 (d, 4H, *m*-Py), 8.32 (d, 4H, *o*-Ph), 7.83 (m, 6H *m,p*-Ph), 6.13 (s, br., 4H, NH_2_ (L)),5.86 (d, *J* = 8.0 Hz, 4H, *m*-Ph (L)), 5.30 (s, br.,2H, OH(L)), 4.09 (s, 2H, -CH-(L)), 3.13 (t., *J* = 2.0 Hz, 4H, -CH_2_-(L)), and 1.90 (d, *J* = 8.0 Hz, 4H, *o*-Ph (L)). UV–Vis (DMSO. nm): λ*_max_*(logε) 598(4.30), 559(4.43), 424(5.67), and 406(4.83). Anal.Calcd.for C_58_H_46_N_8_O_4_Sn: C. 68.99; H. 4.56; N. 8.33. Found: C. 68.95; H. 4.53; and N. 8.30.

Synthesis and spectral characteristics of liner complex *(octopamine)*_2_-Sn(IV)-5,15-diphenyl-10,20-di(4-pyridyl)porphyrin with Cu(II) salts {(Sn(L)_2_P)_2_Cu, **7**}: 7.5 mg (6.8 µmol) of Sn(L)_2_P was mixed with 12 mg (70 µmol) of CuCl_2_·2H_2_O, and dissolved in 5 mL of DMF. The reaction mixture was refluxed for 2 h. Then, the reaction mixture was repeatedly extracted by adding chloroform and water. The chloroform layer was separated, the solvent was evaporated. The product yield was 5.5 mg (71 %). ^1^H NMR (500 MHz, DMSO-d_6_): 9.28. (m, 8H, *β*-H), 9.22 (d, 4H, *β*-H), 9.13 (m, 4H, *o*-Py), 8.34 (d, 4H, *m*-Py), 8.32 (d, 4H, *o*-Ph), 7.87 (m, 6H *m,p*-Ph), 6.13 (s, 2H, NH_2_-L*(→Cu)*), 6.60 (s, 2H, NH_2_-L*(→Cu)*), 5.86 (d, *J* = 8.0 Hz, 4H, *m*-Ph (L)), 5.13 (s, br.,2H, OH(L)), 4.05 (s, 2H, -CH-(L)), 3.10 (t., *J* = 2.0 Hz, 4H, -CH_2_-(L)), 1.89 (d, *J* = 8.0 Hz, 4H, *o*-Ph (L)).UV–Vis (DMSO), λ_max_ (logε): 424(4.98), 558(3.88), 599(3.37). Anal.Calcd.for C_116_H_90_N_16_O_8_Sn_2_Cu: C. 65.17; H. 4.21; and N. 10.47. Found: C. 65.14; H. 4.18; and N. 10.19.

### 2.4. DOSY NMR Spectroscopy

The study of the formation mechanisms and structures of obtained oligomers and polymers in solution was carried out using DOSY NMR spectroscopy. The obtained DOSY NMR spectra were performed on a Bruker Avance III 500 MHz spectrometer with a 5 mm TBI probe head. Temperature stability (25 °C) was controlled using a Bruker unit (BVT-2000) in combination with a Bruker cooling unit (BCU-05) [44]. The 2D DOSY spectra were recorded using the CPMG pulse sequence. The CPMG sequence was used with a diffusion delay of 0.15 s, a total diffusion encoding pulse width of 1.5 ms for each of the 16 gradient amplitudes, 16,384 complex data points were obtained.

To determine the diffusion coefficient values (D), experimental curves of diffusion damping were obtained, which are the dependence of the relative integrated intensity I(I_0_) on the power of the gradient pulse on a logarithmic scale. The resulting curves were approximated by mathematical model including the experimental parameters and the diffusion coefficient (Equation (1)):(1)$$I=I[0]\times \exp(-D\times \mathrm{SQR}(2\times \mathrm{PI}\times \mathrm{gamma}\times \mathrm{Gi}\times \mathrm{LD})\times \left(\mathrm{BD}-\mathrm{LD}/3\right)\times 10^{4})$$ where gamma is the gyromagnetic ratio of the investigated nuclei (4.258·10^3^ Hz/G), Gi is the power of the gradient pulse, the big delta (BD) is 49.9 ms, and the little delta (LD) is 2.8 ms. The data were processed using the qtiplot software.

The approximation by the presented model made it possible to establish the diffusion coefficients values with high accuracy. To determine the molecular weights of the obtained complexes, the signals, and the corresponding solvent coefficients (DMSO-d6 and CDCl_3_ depending on the system) were chosen as standards. The diffusion coefficients of the research objects based on porphyrins were measured using the CPMG method (Carr-Purcell-Meiboom-Gill) [45,46,47,48]. The high measurement accuracy (±0.20·10^−10^ m^2^/s) confirms that the sensitivity of the DOSY CPMG method is sufficient for the experimental separation of structures of various sizes, shapes, and molecular weights.

To interpret and visually demonstrate the results obtained, the method of graphical analysis was applied [33,49,50,51]. The method is based on the interdependence of the translational diffusion coefficient and mass in accordance with the Einstein–Smoluchowski relation. It was shown that the ratio of the diffusion coefficients for two different molecules (Di/Dj) is inversely proportional to the square or cubic root of the ratio of their molecular weights (Mj/Mi) for rod-shaped and spherical molecules, respectively [50,52,53,54]:(2)$$\sqrt[{2}]{\frac{M_{j}}{M_{i}}}\geq \frac{D_{i}}{D_{j}}\geq \sqrt[{3}]{\frac{M_{j}}{M_{i}}}$$

In accordance with Equation (2), the theoretical values of the diffusion coefficients were obtained from the molecular weight and the diffusion coefficient values of the solvent. Experimentally determined values of the diffusion coefficient for the objects of study were plotted on the obtained theoretical curves. The experimental values (points on the graphs) are within the range between the theoretical curves, which confirm that the diffusion coefficients correspond to the molecular weights of the proposed structures. All the obtained values of the diffusion coefficients and the molecular weights of the formed supramolecular structures are given in Table 1, Table 2, Table 3 and Table 4. 

### 2.5. Photoresistance and Thermal Stability Study Technique

The study of the photo-oxidation process was carried out by irradiating dilute solutions of porphyrins for 60–75 min using a UV lamp (λ = 415 nm, Ev = 1790 ± 30 lx). Fluorescence spectra were recorded at λ_exe_ = 420 nm, the width of the excitation and emission slit is 5 nm. The photo-oxidation process was studied in dimethyl sulfoxide (DMSO) medium. The changes in the UV-Vis and fluorescence spectra were recorded every 15 min. The photodestruction degree (η, %) was calculated from the decrease in the optical density of the Soret band. The photo-oxidation constant (k, min^−1^) was obtained graphically using the dependence lnC_porph_. = f(τ). The half-life was calculated by Equation(3):(3)$$\tau_{1/2}=\frac{ln2}{k}$$

The thermal stability of polymers in high-boiling DMSO (189 °C) was also controlled using UV-Vis and fluorescence spectra. The spectra were recorded at certain temperature intervals.

### 2.6. Determination of the Porosity of Hybrid Sn(IV)-Porphyrin Polymers

The porosity of the samples was studied on a NOVAtouch NT LX specific surface and porosity analyzer. Pore distribution analysis was performed by the Barret–Joyner–Halenda (BJH) method. Before measurements, the samples were degassed for 1 h at 523 K. Liquid nitrogen was used as the adsorbate (bath temperature 77.35 K). The samples were heated to 100.0 °C at a heating rate of 10 °C/min, then kept at this temperature for 180 min. 

## 3. Results and Discussion

### 3.1. Structures of the Sn(IV)-Dipyridylporphyrin Based Systems

In this work, to create supramolecular porphyrin arrays, the chelation reaction of diaxial complexes of the Sn(IV)-dipyridylporphyrin with Cu(II) and Pd(II) cations wereused. *Dihydroxy*-5,15-diphenyl-10,20-di(4-pyridyl)porphyrinate Sn(IV) **1** and its diaxial complex with octopamine **6** were chosen as monomeric tetrapyrrole units for the preparation of porphyrin oligomers and polymers. To prove the structure of the synthesized complexes, one- (^1^H NMR, ^13^C NMR) and two-dimensional (^1^H-^1^H COSY and ^1^H-^13^C HSQC) NMR spectroscopy was used. The spectra obtained with the correlation of the protons and carbon atoms signals are shown in Appendix A.

The complex **6** formation is accompanied by a significant shift of the proton signals of the octopamine phenyl ring in ^1^H NMR spectrum, while the porphyrin bands remain almost unchanged (Figure 1). The *m*-Ph and *o*-Ph protons of octopamine shift from 7.86 to 5.86 ppm and from 8.38 to 1.90 ppm, respectively (Figure 1). It should also be noted that the OH-protons signals of octopamine disappear in the region of 9.51 ppm (Figure 1), which indicates binding to Sn(IV)porphyrin precisely through the OH-group of the octopamine phenyl fragment.

The formation of linear dimer **7**, which occurs due to the binding of octopamines in molecules **6** by Cu(II) salts, leads to splitting of NH_2_ proton signals into ^1^H NMR (6.13 and 6.60 ppm), and a decrease in the integrated intensity of OH proton signals in the 5.13 ppm region (Figure 2).

In addition to the synthesis of the compounds presented in Section 2.3, the linear oligomers **5** based on Pd-salts were obtained. The solution obtained by boiling Sn(OH)_2_Pand salt (C_6_H_5_CN)_2_PdCl_2_in a ratio of 2:1 in DMF was studied using DOSY NMR spectroscopy. The study showed (Table 1, Figure 3A) that in the course of this reaction, a cyclic tetrameric product is formed as the main product, although the dimer and monomer are also presented in the reaction mixture, but in a very small amount. The reaction course is determined by the Pd-salt structure having the form of a *cis*-isomer. Therefore, the task was to purposefully cyclize the tetramer structure based on dihydroxy-Sn(IV)-5,15-diphenyl-10,20-di(4-pyridyl)porphyrin.

To obtain the cyclic tetramer **2** structure, the ratio of reagents was changed to 1/1. The resulting reaction mixture (synthesis of tetramer **2** (Sn(OH)_2_P)_4_(PdCl_2_)_4_according to the method presented in Section 2.3) was also investigated using DOSY NMR spectroscopy. In the reaction mixture, in addition to the monomeric form of the cyclic tetramer **2**, there is also dimeric form (Figure 3B, Table 2). In addition, the solution also contains structures consisting of three and four cyclic tetramers **2** (Figure 3B, Table 2).

The formation of the tetramer **2** from the Sn(OH)_2_P **1** is accompanied by the displacement of *β*-pyrrole and *o*-Py protons in ^1^H NMR spectrum (Figure 4) [51]. The signals of *β*-pyrrole protons at 9.28 and 9.22 ppm are shifted to 9.20 and 9.16 ppm, respectively. In turn, the *o*-Ph protons signals are downfield shifted to 9.30 ppm. The UV-Vis spectrum of the system shows blue shift (2–5 nm) of the Q_x_- and Soret bands, and the appearance of a band at 628 nm, which corresponds to binding of monomeric fragments of the **1** with Pd(II) salts.

For the cyclic tetramer with octopamine **3** {(Sn(L)_2_P)_4_(PdCl_2_)_4_}, the diffusion coefficient was obtained (Table 3) from the graph of its dependence on the molecular weight of the complex. It turned out that the monomeric form of compound **3** exists in the reaction mixture. No complexes with other structures were found in the reaction mixture.

The results obtained indicate that the structures formed in solution based on the tetramer **2** (Sn(OH)_2_P)_4_(PdCl_2_)_4_, consisting of 2–4 units, are apparently linked by weak hydrogen bonds via the axial OH groups. This assumption is supported by the absence of associates consisting of several units in the solution when the OH group is replaced by octopamine (Table 3).

Next, the polymer structure **4** ((Sn(L-Cu-)_2_P)_4_(PdCl_2_)_4_)_n_ was synthesized by binding of compound **3** (Sn(L)_2_P)_4_(PdCl_2_)_4_with copper(II) salts. The study of the reaction mixture using DOSY NMR spectroscopy showed the formation in the solution of structures consisting of 2–10 monomer units of the tetramer (Sn(L)_2_P)_4_(PdCl_2_)_4_(Table 4, Figure 5).

The synthesized structures 1, 2, and 4 were also characterized by IR spectroscopy. IR spectroscopy data are presented in the Appendix A.

The geometrical optimization of experimentally obtained diaxial complexes based on Sn(IV)-diphenyldipyridylporphyrines and their associates (porphyrin oligomers) of various compositions was carried out by the density functional theory (DFT) method, the B3LYP functional and the combination of the 6-31G basis set for O, H, C, N and def2-TZVP for Sn. Geometric and energy characteristics of optimized molecules and polymers are shown in Table 5, Table 6 and Table 7.

The structure optimization by quantum chemical calculations showed that the partial transfer of an electron pair of the N_p_ atom to the empty Sn-atom orbitals (LPN → LP*Sn) occurs in Sn(L)_2_P **6** with two axial ligands. According to the NBO analysis, the stabilization energy value of the Sn-N_p_ bond is 195.2 kJ/mol, and the corresponding charge transfer (q_st_) value is 0.455 e (Table 5).

At the same time, there is the transfer of an electron pair from the oxygen atom of the ligand to an empty orbital of the Sn atom according to the LPO → LP*Sn type. The Sn-O bond stabilization energy value is 230.4 kJ/mol, and the corresponding charge transfer value is 0.608 e (Table 5).

The <Sn-O-L angle value given in Table 5 shows the tilt angle of the aromatic ring of the ligand relative to the porphyrin ring, which allows to evaluate the conjugation degree of the ligand with the macrocycle (π-π interaction). Thus, the conjugation degree of the ligand L with the macrocycle is less than with the OH group.

We also performed the geometry optimization of the monomeric unit in the chain polymer (M^2+^=Cu^2+^) by the DFT method with the CAM-B3LYP/3-21G kit. The geometry selected bond lengths and angles of the optimized molecules are shown in Table 6.

The analysis of the obtained results showed that the inclusion of the monomer unit based on Cu(II) in the polymer chain leads to an increase in the bond length r(Sn-O) compared to the initial complex Sn(L)_2_P (Table 5 and Table 6). The possible reason for this may be an increase in the binding energy of the ligand with Cu(II). Additionally, with increasing the bond length r(Sn-O), the decrease in the charge transfer value (q_st_) from the oxygen atom of the ligand to the tin atom of the complex (LPO → LP*Sn) is observed. Thus, the binding energy of the ligand to the macrocycle decreases compared to the initial Sn(L)_2_P, and the inclination angle of the ligand relative to the porphyrin ring (<Sn-O-L) decreases, which leads to а greater conjugation of the ligand with the macrocycle.

Table 7 and Appendix A present the data of the 2D tetramers optimizing structures based on Sn(L)_2_P porphyrin and Pd(II) cations. According to the above quantum chemical calculations, the cell cavity of the polymer based on Sn(L)_2_P is a parallelogram with dimensions of 18.12 × 19.23 × 19.01 Å, and the cell volume of 5660 Å^3^, respectively, which is in good agreement with the experimental data.

### 3.2. Powder X-ray Diffraction Studies and Thermogravimetric Analysis

According to the powder X-ray diffraction studies, the Sn(OH)_2_P **1** and the triad **6** have crystalline structure. Further complication of the studied systems structure due to the cyclization or linear dimerization leads to a decrease in the crystallinity of the samples (Figure 6). However, oligomers **2**, **4**, and **7** are not amorphous; they still retain their crystalline structure, which indicates ordered assembly of monomeric units in their composition.

The TGA data was given in the Appendix A. The first mass loss peak, which starts at 79.9 *°*С, corresponds to the evaporation of solvent molecules (DMF). The second peak of weight loss (t = 296.1–361.2 *°*С) corresponds to the destruction of octopamine fragments, which leads to the breakdown of the polymer **4** framework. The next stage of degradation begins at a temperature of 466.9 *°*С. This process is associated with the destruction of tetrameric structures, that is, the destruction of Py-Pd-Py bonds, into monomeric fragments.

### 3.3. Determination of the Porosity of the Hybrid Sn(IV)-Porphyrin Polymers

Based on the obtained data on the pore size distribution, it can be concluded that the synthesized polymer structures are predominantly materials with micro- and meso-pores (Table 8, Appendix A). According to the adsorption data (Table 8), the pore radius of monomers **1**, **6,** and their cyclic tetramers **2**, **3** practically do not differ. However, it should be noted that the octopamine introduction (compounds **3** and **6**) contributes to the increase in the pore radius compared to dihydroxy derivatives **1** and **2**. It can also be concluded that the tetramers **2**, **3,** and **4** formation is accompanied by the significant increase in the pore volume and surface area compared to **1**, **6,** and **7**. However, the pore volume and the surface area values of the obtained polymer structure **4** are much lower than for tetramer **2** (Table 8, Appendix A). This is quite expected and explained by a denser arrangement of molecules (Sn(L)_2_P)_4_(PdCl_2_)_4_due to self-assembly by Cu(II) salts in polymer **4** than in the free form. The decrease in the surface area and pore volume is also observed for compound **7** obtained by bonding two molecules of Sn(L)_2_P (compound **6**) with Cu(II) cations.

Figure 7 shows the adsorption and desorption isotherms of the synthesized compounds. Based on the type of isotherms and data in Table 8 and Appendix A, it could be concluded that the compounds are predominantly mesoporous materials with the cylindrical pore shape. Isotherms A–D in Figure 7 have a narrow hysteresis loop, which indicates the presence of narrow mesopores having a shape close to conical in the compounds **1**, **2**, **6,** and **7**.

An increase in the pore radius in the compounds explains the partial filling of pores before the start of the experiment and their emptying during desorption. The reversibility of the gas adsorption–desorption process (adsorbed nitrogen molecules do not linger in the pores of polymers during desorption) testifies in favor of the possibility of their practical application as molecular containers for storage, transportation, and release of reactive gases.

### 3.4. Photo- and Thermal Stability of Obtained Poly-Sn(IV)Porphyrin Systems in Solution

In the study of the effect of environmental conditions on the spectral-fluorescent properties of the synthesized Sn(IV)porphyrin systems, it was found that the construction complication of the (Sn(OH)_2_P)_4_(PdCl_2_)_4_ building block leads to an increase in the photostability of the investigated samples (Table 9). The values of the photo-degradation degrees, the photo-oxidation constants of the tetramer with octopamine extra-ligands **3** (Sn(L)_2_P)_4_(PdCl_2_)_4_, and the polymer **4** ((Sn(L-Cu)_2_P)_4_(PdCl_2_)_4_)_n_decrease by about 3–4 times compared with tetramer **2** (Sn(OH)_2_P)_4_(PdCl_2_)_4_. The half-life values, on the contrary, increase, which also indicates an increase in the stability of compounds **3** and **4** compared to tetramer **2**.

In contrast, the UV irradiation of the compounds **1-6** is accompanied by an increase in their fluorescence on average by 10%. Figure 8 shows the spectral changes of (Sn(L)_2_P)_4_(PdCl_2_)_4_observed during 75 min UV photo-irradiation.

The thermal stability of the compounds **1-6** was assessed by the change in the UV-Vis and fluorescence spectra when they were boiled in DMSO. In the UV-Vis spectrum, when the temperature of the solution reaches 112 °C, there is a slight decrease in optical density with the blue shift of the Soret band from 425 nm to 424 nm (Figure 9 on the left), while in the fluorescence spectra, on the contrary, an increase in the intensity of the bands was recorded (Figure 9 on the right). The further increase in temperature to 140 °C again leads to an increase in optical density in the UV-Vis spectrum with an already red shift of the Soret band to 427 nm, while a decrease in fluorescence properties is recorded in the fluorescence spectra.

The tetramer with octopamine **3** turned out to be more unstable than the tetramer **2**; however, the trend of spectral changes generally remains. Figure 10 shows the changes in the UV-Vis spectrum of (Sn(L)_2_P)_4_(PdCl_2_)_4_during the variation of the solution temperature in DMSO. First, there is a slight drop in the optical density of the Soret band when the solution temperature reaches 50 °C; however, a further increase in temperature to 110 °C leads to an increase in the optical density. The subsequent increase in temperature in the 110–183 °C range again leads to a decrease in the maxima of the B and Q_x_-bands.

In the fluorescence spectra of the compound **3** (Sn(L)_2_P)_4_(PdCl_2_)_4_ with an increase intemperature from 23 °Cto 110 °C, there is the tendency to an increase the fluorescence (Figure 11). Afurther increase intemperature to 183 °C again leads to a decrease inthe band maxima intensity in the fluorescence spectra (Figure 11). Recall that the similar trend is observed for unsubstituted tetramer **2** (Figure 9 on the right).

In contrast to the compounds considered above, in the UV-Vis spectrum of polymer **4**, the decrease in the optical density of both the Soret band and the Q_x_-bands is observed with an increase in the temperature from 23 °Cto 187 °C. However, it should be noted that the Soret band first shifts in the short-wavelength region by 3 nm with increasing temperature, and one returns again to 427 nm when the temperature reaches 187 °C.

The fluorescence spectra of the polymer **4** ((Sn(L-Cu)_2_P)_4_(PdCl_2_)_4_)_n_ (Figure 12) show the same trend as for the objects described above. In the temperature range of 23–110 °C, an increase in the fluorescence was recorded, and a further increase in temperature again leads to its quenching. The fluorescence of compounds **3** and **4** is higher at the solution temperature of 183 °C compared to the room temperature solution (Figure 11 and Figure 12).

In [55], the photophysics of a conjugated metal-bound dimeric porphyrin with one palladium cation was studied spectroscopically and using quantum chemical calculations. The influence of the conformation on the ground and first singlet excited states has been studied. Absorption spectra and quantum chemical calculations revealed two different conformations of the dimer. The calculated value of the torsion energy barrier is ΔЕ = 3.4 kJ·mol^−1^.

The torsion angle of porphyrins relative to each other in the dimer determines the distribution of electronic transitions in the molecule, and hence its UV-Vis and emission spectra. As the rotation angle decreases, the conjugation of porphyrin fragments with each other decreases, and molecular orbitals are localized only on one porphyrin fragment. Moreover, complete charge transfer from ligand to ligand can take place. Studies of the temperature dependence of the fluorescence spectrum of the dimer showed the fundamental possibility of using it as a contactless luminescent temperature sensor.

Most likely, a similar phenomenon is also observed in the case of porphyrin oligomers with palladium cations (tetramers and a polymer) studied in this work. Quantum chemical studies to confirm the presence of various conformers of porphyrin oligomers were not included in this study. However, it can be observed that, in the viewing temperature range from 23 °C to 110 °C, the torsion angles of rotation in oligomers also change from 90 °C to 180 °C, which leads to the ignition of their fluorescence. An increase at a temperature of 110 °C leads to a sharp reduction in the photophysical behavior of porphyrin arrays, which may be higher than the complete detection of energy barrier overcoming with the formation of a conformer characterized by an inversely proportional distribution of porphyrin fragments. Due to the presence in porphyrin arrays (tetramers and polymers) of several binding sites through palladium cations, the fluorescence temperature dependence of the compounds described in this work is even more pronounced than in the case of a dimer [55].

## 4. Conclusions

New methods for preparation of hybrid metalloporphyrin polymers based on palladium (II) and copper (II) cations and diaxial complexes of dipyridyl-substituted Sn(IV)porphyrin were developed. The structure and the formation mechanism of the obtained hydrophobic Sn(IV)-porphyrin oligomers and polymers in the solution were established, and their resistance to the UV radiation and changes in solution temperature werestudied. It was shown that the obtained polyporphyrin nanostructures are porous materials with predominance of cylindrical mesopores. Density functional theory (DFT) was used to determine the sizes of unit cells in the polyporphyrin tubular structures. The results obtained can be used to create highly porous materials for separation, storage, transportation, and controlled release of substrates of different nature, including highly volatile, explosive, and toxic gases.

## Figures and Tables

**Figure 1 polymers-15-01055-f001:**
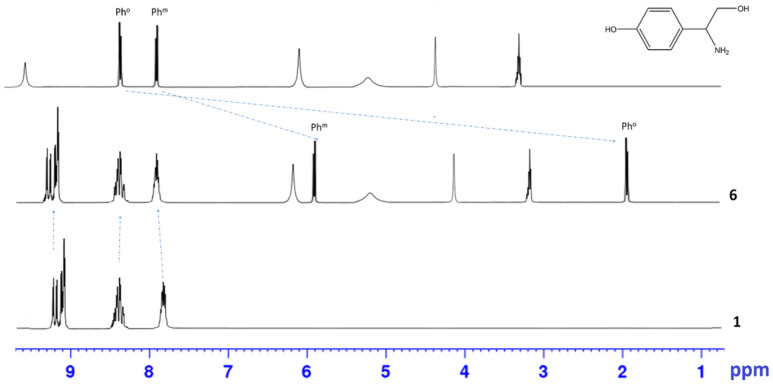
^1^Н NMR-spectrum of the Sn(OH)_2_P **1**, Sn(L)_2_P **6,** and octopamine in DMSO-*d_6_*.

**Figure 2 polymers-15-01055-f002:**
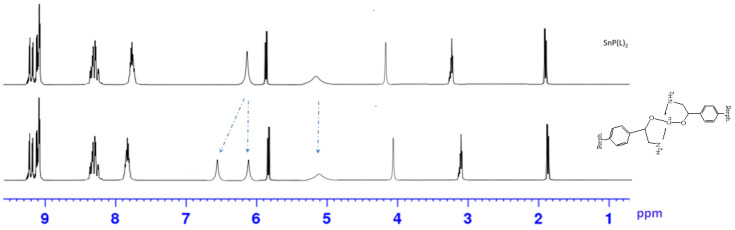
^1^H NMR-spectrum of the Sn(L)_2_P **6** and (Sn(L)_2_P)_2_Cu **7**.

**Figure 3 polymers-15-01055-f003:**
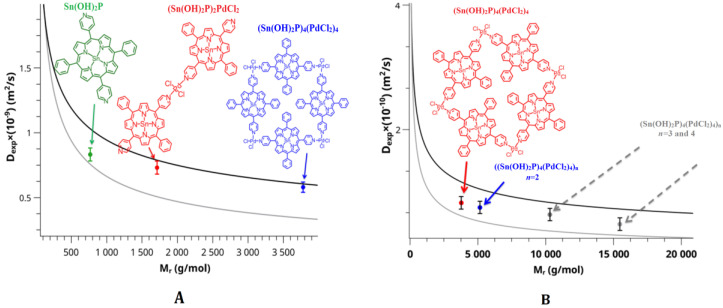
Graphical analysis of self-diffusion coefficients versus molecular weight from DOSY NMR spectroscopy. (**A**)The green dot is the monomeric structure of Sn(OH)_2_P **1**, the red dot is the structure of complex **5**, the blue dot is the structure of complex **2.** (**B**) The red dot is the monomeric structure of the (Sn(OH)_2_P)_4_(PdCl_2_)_4_ complex **2**, the blue dot is the dimeric structure of the complex **2**, two additional gray dots correspond to the trimeric and tetrameric structures of the complex **2**, the presence of which is also likely.

**Figure 4 polymers-15-01055-f004:**
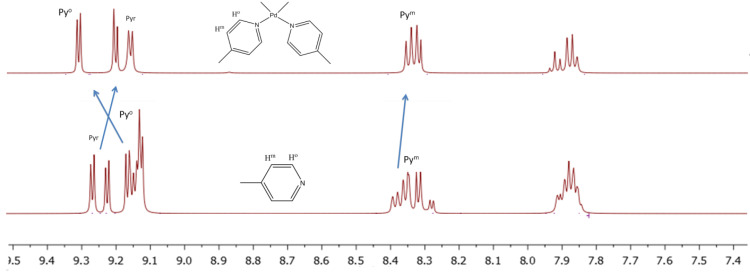
^1^H NMR-spectrum of the cyclic tetramer **2**{(Sn(OH)_2_P)_4_(PdCl_2_)_4_} and Sn(OH)_2_P**1**.

**Figure 5 polymers-15-01055-f005:**
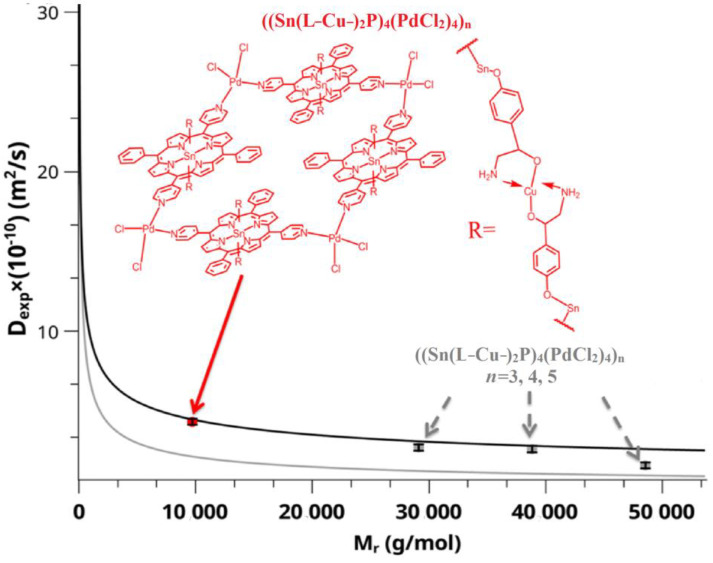
Graphical analysis of self-diffusion coefficients versus molecular weight from DOSY NMR spectroscopy. The red dot is the dimeric structure of the complex ((Sn(L-Cu-)_2_P)_4_(PdCl_2_)_4_)_n_, the gray dots are the probable structure of the complex formed from several monomer units.

**Figure 6 polymers-15-01055-f006:**
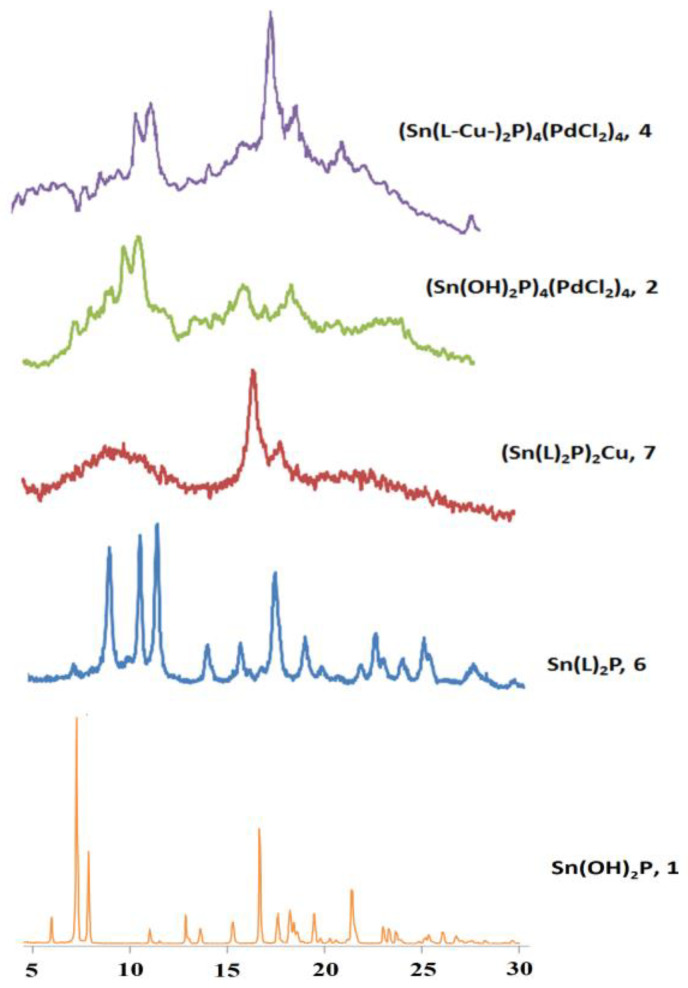
Powder X-ray diffraction (PXRD) of Sn(L)_2_P **6, (**Sn(L)_2_P)_2_Cu **7,** (Sn(OH)_2_P)_4_(PdCl_2_)_4_**2**_,_and((Sn(L-Cu-)_2_P)_4_(PdCl_2_)_4_)_n_**4**.

**Figure 7 polymers-15-01055-f007:**
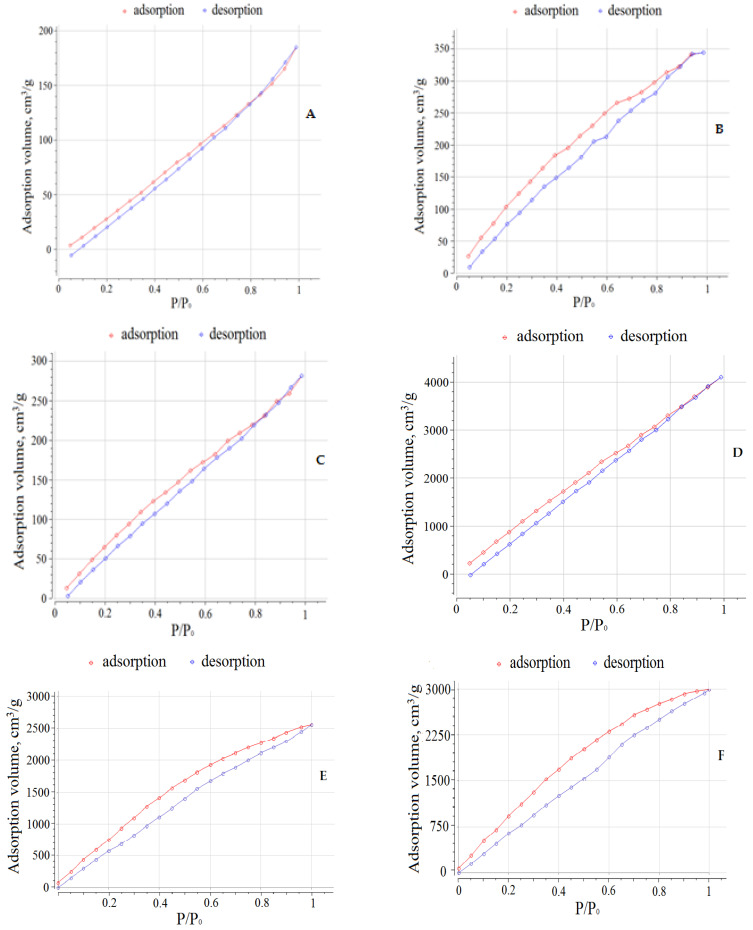
Adsorption and desorption isotherms of the compounds **1** (**A**), **6** (**B**), **7** (**C**), **2** (**D**), **3** (**E**), and **4** (**F**).

**Figure 8 polymers-15-01055-f008:**
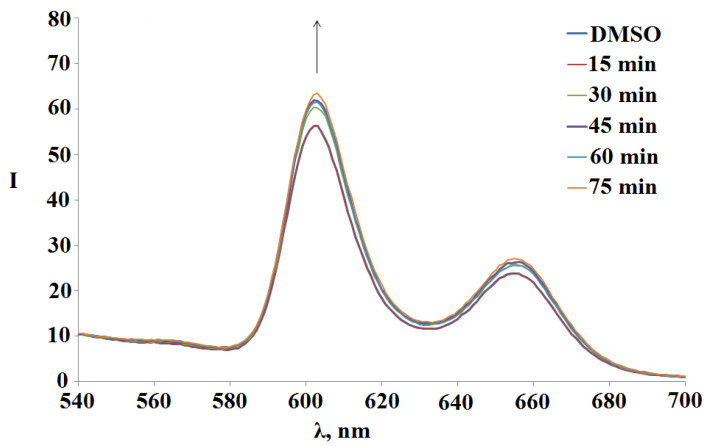
Changes in the fluorescence spectra of tetramer with octopamine **3** (Sn(L)_2_)_4_(PdCl_2_)_4_ during 75 min UV photo-irradiation in DMSO.

**Figure 9 polymers-15-01055-f009:**
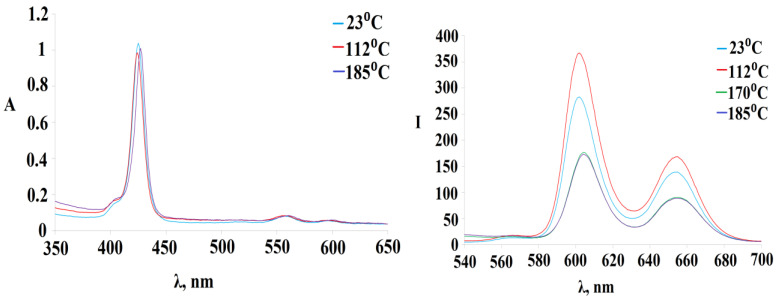
Dependence of UV-Vis spectrum and fluorescence spectra of (Sn(OH)_2_P)_4_(PdCl_2_)_4_ on the solution temperature.

**Figure 10 polymers-15-01055-f010:**
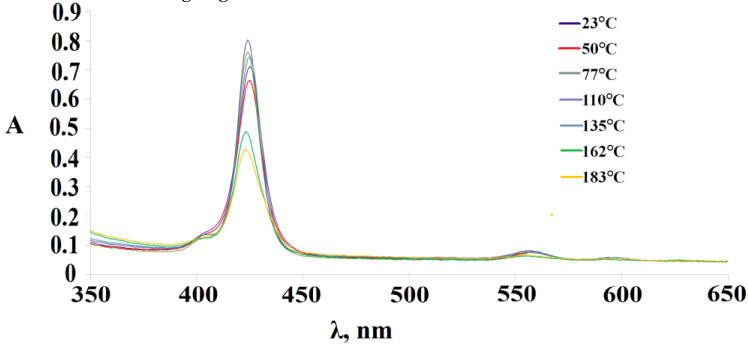
Changes in the UV-vis spectra of the (Sn(L)_2_P)_4_(PdCl_2_)_4_(compound **3**) when the temperature of the DMSO solution varies from 23 °Cto 183 °C.

**Figure 11 polymers-15-01055-f011:**
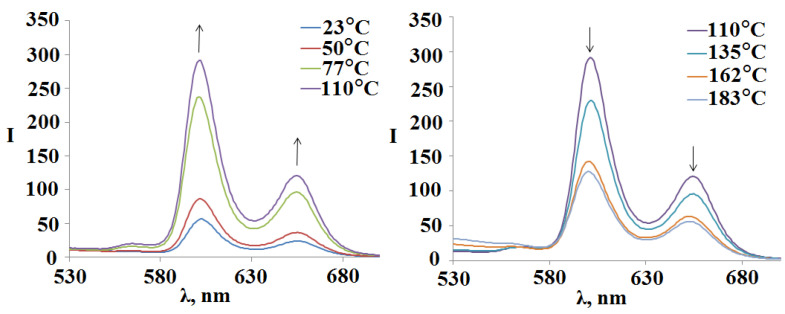
Change in the fluorescence spectra of the compound **3** (Sn(L)2P)4(PdCl_2_)4 when the temperature of the DMSO solution varies from 23 °Cto183 °C.

**Figure 12 polymers-15-01055-f012:**
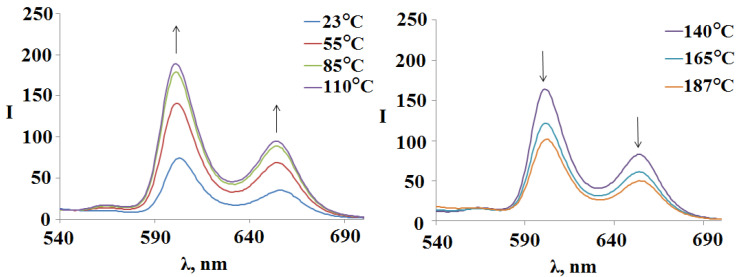
Change in the fluorescence spectra of the polymer **4** ((Sn(L-Cu)_2_P)_4_(PdCl_2_)_4_)_n_ when the temperature of the DMSO solution varies from 23 °Cto 187 °C.

**Table 1 polymers-15-01055-t001:** Diffusion coefficients of compounds from the reaction mixture formed during the preparation of the porphyrin arrays with different numbers of structural units.

Name	Number of Monomer Units	D_exp_ (×10^−9^) (m^2^/s)	Molecular Weight (g/mol)
Sn(OH)_2_P **1**	Monomer	0.83	767.42
(Sn(OH)_2_P)_2_PdCl_2_ **5**	Dimer	0.73	1712.26
(Sn(OH)_2_P)_4_(PdCl_2_)_4_ **2**	Tetramer	0.58	3779.36

**Table 2 polymers-15-01055-t002:** Summary results of DOSY NMR experiments, for the object of study **2** with the values of the molecular masses of the complexes, and the diffusion coefficients determined from the experiments. The molecular weight and diffusion coefficient of dimethyl sulfoxide-d6 (DMSO-d6) are used as a reference in the calculations.

Name	Number of Monomer Units	D_exp_ (×10^−10^) (m^2^/s)	Molecular Weight (g/mol)
DMSO	Reference	4.13	84.17
(Sn(OH)_2_P)_4_(PdCl_2_)_4_ **2**	Mono-	0.83	3779.36
	Di-	0.75	7558.72
	Three-	0.64	11,338.08
	Tetra-	0.56	34,014.24

**Table 3 polymers-15-01055-t003:** Summary results of DOSY NMR experiments, the molecular weight and the diffusion coefficient values for the cyclic tetramer with octopamine **3**. The molecular weight and diffusion coefficient of DMSO-d6 are used as reference in the calculations.

Name	Number of Monomer Units	D_exp_ (×10^−10^) (m^2^/s)	Molecular Weight (g/mol)
DMSO	Reference	5.21	84.17
(Sn(L)_2_P)_4_(PdCl_2_)_4_ **3**	Mono-	0.86	5152.345

**Table 4 polymers-15-01055-t004:** Summary results of DOSY NMR experiments, the molecular weight and the diffusion coefficient values for the research object **4**. The molecular weight and diffusion coefficient of deuterium chloroform (CDCl_3_) are used as reference in the calculations.

Name	Number of Monomer Units (*n*)	D_exp_ (×10^−10^) (m^2^/s)	Molecular Weight (g/mol)
CDCl_3_	Reference	19.4	119.38
((Sn(L-Cu)_2_P)_4_(PdCl_2_)_4_)_n_ **4**	Di-	4.32	9704.69
	Hexa-	2.71	29,114.07
	Octa-	2.61	38,818.76
	Deca-	1.59	48,523.45

**Table 5 polymers-15-01055-t005:** Geometric parameters of the diaxial complexes based on Sn(IV)-diphenyldipyridylporphyrines.

Name	*r*(Sn-O), Å	*E*_st_(Sn-O), kJ/mol	*q*_st_,*e*	*r*(Sn-N_p_),Å	*E*_st_(Sn-Np), kJ/mol	*q*_st_,*e*	<L-O-O-L, °	<Sn-O-L, °
Sn(OH)_2_P **1**	1.998	451.9	0.481	2.101	397.5	0.521	78.3	116.3
Sn(L)_2_P **6**	2.033	230.4	0.608	2.098	195.2	0.455	129.0	132.1

**Table 6 polymers-15-01055-t006:** Geometrical parameters of the monomer unit in the chain Cu(II) polymer based on Sn(L)_2_P.

Name	*r*(Sn-O),Å	*E*_st_(Sn-O)kJ/mol	*q*_st_,*e*	*r*(Sn-N_p_),Å	*E*_st_(Sn-Np), kJ/mol	*q*_st_,*e*	<Sn-O-L,°	*r*(Sn-Sn),Å
(Sn(L)_2_P)_2_Cu **7**	2.038	207.1	0.446	2.091	242.9	0.678	122.5	18.124

**Table 7 polymers-15-01055-t007:** Geometrical parameters of the studied compounds obtained by quantum chemical calculations using the method DFT/B3LYP/3–21G.

Name Parameters	(Sn(OH)_2_P)_4_(PdCl_2_)_4_ 2	(Sn(L)_2_P)_4_(PdCl_2_)_4_ 3
*r*(Sn-O), Ǻ	2.014	2.011
*r*(Sn-N), Ǻ	2.112	2.108
<Sn-O-L, °	114.1	114.6
*d*(Sn…Sn) *X*-axis *	18.98	19.23
*d*(Sn…Sn) *Y*-axis **	18.84	19.01

*d*(Sn…Sn) *X*-axis *- distance between tin atoms of porphyrins along the X axis. *d*(Sn…Sn) *Y*-axis; **- distance between tin atoms of porphyrins along the Y axis.

**Table 8 polymers-15-01055-t008:** Average pore radius ®, pore volume (V), and surface area (S) obtained by the BJH and BET methods.

Name	Adsorption			Desorption		
	S_,_m^2^/g	V, cm^3^/г	r, nm	S_,_m^2^/g	V, cm^3^/g	r, nm
BJH						
Sn(OH)_2_P **1**	154.686	0.27	1.68	192.71	0.31	1.53
Sn(L)_2_P **6**	257.868	0.38	1.88	334.319	0.48	2.15
(Sn(L)_2_P)_2_Cu **7**	214.574	0.35	2.11	238.272	0.39	1.91
(Sn(OH)_2_P)_4_(PdCl_2_)_4,_ **2**	3420.48	5.41	1.69	3699.60	5.89	1.70
(Sn(L)_2_P)_4_(PdCl_2_)_4,_ **3**	2340.82	3.99	1.89	2865.23	4.59	1.89
((Sn(L-Cu)_2_P)_4_(PdCl_2_)_4_)_n,_ **4**	2694.52	4.28	1.90	3190.07	4.85	1.90
BET						
Sn(OH)_2_P **1**	158.278	0.26	1.68	197.177	0.30	1.53
Sn(L)_2_P **6**	263.959	0.37	1.88	342.071	0.47	2.15
(Sn(L)_2_P)_2_Cu **7**	219.609	0.35	2.11	243.848	0.38	1.91
(Sn(OH)_2_P)_4_(PdCl_2_)_4,_ **2**	3499.82	5.30	1.69	3786.12	5.77	1.70
(Sn(L)_2_P)_4_(PdCl_2_)_4,_ **3**	2462.23	3.89	1.89	2932.47	4.49	1.70
((Sn(L-Cu)_2_P)_4_(PdCl_2_)_4_)_n,_ **4**	2757.43	4.19	1.90	3183.41	4.75	1.90

**Table 9 polymers-15-01055-t009:** Experimental data of the objects of study after a session of UV light irradiation after 75 min (λ = 415 nm) in DMSO.

Name	η, %	k, min^−1^	τ_1/2_, min
Sn(OH)_2_P **1**	15	0.0021	237
(Sn(OH)_2_P)_4_(PdCl_2_)_4_ **2**	11	0.0015	462
(Sn(L)_2_P)_4_(PdCl_2_)_4_ **3**	4	0.0004	1733
((Sn(L-Cu)_2_P)_4_(PdCl_2_)_4_)_n_ **4**	3	0.0005	1386
Sn(L)_2_P **6**	7	0.0009	893

## Data Availability

Not applicable.

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
