# Peer review of "Synthesis and Design of Hybrid Metalloporphyrin Polymers Based on Palladium (II) and Copper (II) Cations and Axial Complexes of Pyridyl-Substituted Sn(IV)Porphyrins with Octopamine"

_polymers, 2023, doi:10.3390/polym15041055_

Round 1
Reviewer 1 Report
The manuscript from A. E. Likhonina and co-worker describes the synthesis and characterize of new supramolecular porphyrin oligomers and polymers by use Sn porphyrin diaxial complexes with Cu and Pd cations. The author performs a series of physical characterization techniques such as NMR, PXRD, UV-Vis, fluorescent spectra and porosity analyzer. However, the following issues should be emphasized and elaborated in more detail.
1. To facilitate comparison, The author should add the NMR spectra of 1 to the Figure
2. The author stated that 1 and 6 have a crystalline structure. However, I haven’t found the PXRD patterns of 1.
3. The author claimed that the compounds are predominantly mesoporous materials with a cylindrical pore shape. However, it’s difficult to judge the adsorption type. Moreover, “narrow hysteresis loop” hard to convince readers.
4. Why the author hasn’t performed the TGA or PXRD to assess the stability of compounds?
5. Could the author provide a reasonable explanation for the increase in the intensity of the band at 112℃ in the fluorescence spectrum of (Sn(OH)2P)4(PdCl2)4?
Author Response
Responses to your comments are attached as a file.
Thanks a lot for the comments; they have significantly improved our article.
We are ready to provide additional information if necessary.
Sincerely,
authors.

Reviewer 2 Report
1. Review all graphics, subtitles are small, ariel, no pattern. This seems irrelevant but it organizes the work for the reader.
2. The quality of the figures are badly, such as Fig.7.
3. why not test the IR and discuss it? It also should be tested the TGA and DSC.
4. Please also test the BET for checking the porosity.
5. “Different functional substituents, the type of organic fragments binding, the metal nature, and the length of the linking fragments provide a variety of framework structures types and areas of their potential application.” This part should be updated the current refs, such as J. Colloid. Interf. Sci, 2022, 621, 180-194; Inorganics, 10(2022) 202; Micropor. Mesopor. Mat, 341(2022) 112098 and J. Solid State Chem. 318(2023) 123713.
Author Response

(The authors gave the same response as above.)

Round 2
Reviewer 1 Report
The manuscript can be accepted in present form.
Reviewer 2 Report
accepted.